# Ab Initio Study of Structural, Electronic, and Thermal Properties of Pt/Pd-Based Alloys

**Louise Magdalene Botha** [1,*]☺, **Cecil Naphtaly Moro Ouma** [1]☺, **Kingsley Onyebuchi Obodo** [1], **Dmitri Georgievich Bessarabov** [1,*]☺, **Denis Lvovich Sharypin** [2], **Pyotr Sergeevich Varyushin** [2] **and Elizaveta Ivanovna Plastinina** [2]

1 HySA-Infrastructure, Faculty of Engineering, North-West University, Private Bag X6001, Potchefstroom 2520, South Africa; moronaphtaly84@gmail.com (C.N.M.O.); obodokingsleyo@gmail.com (K.O.O.)
2 PJSC MMC Norilsk Nickel, 15, 1st Krasnogvardeysky Drive, 123100 Moscow, Russia; sharypindl@nornik.ru (D.L.S.); varyushinps@nornik.ru (P.S.V.); plastininaei@nornik.ru (E.I.P.)
* Correspondence: louise.botha@nwu.ac.za (L.M.B.); dmitri.bessarabov@nwu.ac.za (D.G.B.)

**Abstract:** Alloys are beneficial in numerous applications since they combine the desirable properties of different metals. In this regard, Pt/Pd alloys have been investigated as a replacement for Pt, which is the standard catalyst used in various catalytic processes. However, there are still gaps in our understanding of the structural, mechanical, and thermodynamic properties of Pt/Pd alloys. This study was conducted using density functional theory (DFT) calculations to investigate the electronic, elasticity, mechanical, and thermodynamic properties of Pt/Pd alloys and compared them to pristine Pt and Pd structures. The results indicate that the considered Pt/Pd alloy structures, PtPd$_3$, PtPd, Pt$_3$Pd, and Pt$_7$Pd, are energetically favourable based on their formation energies. These structures also satisfy Born's stability criteria and are elastically stable. The phonon density of states showed that the considered Pt/Pd alloy structures are dynamically stable, with no imaginary modes present. Additionally, the Pt atom dominates at lower frequencies, while the Pd atom dominates at higher frequencies, as seen in the phonon band structure. The electronic density of states revealed that the considered Pt/Pd alloy structures have a metallic character and are non-magnetic. These findings contribute to a better understanding of the properties and stability of Pt/Pd alloy structures that are relevant in various fields, including materials science and catalysis.

**Keywords:** bimetallic platinum palladium; density functional theory; phonon dynamics; phonon band structure; elastic properties; phase stability

## 1. Introduction

In response to critical concerns, such as climate change, energy security, and decreasing dependency on fossil fuels, efforts have aimed towards the development of clean energy solutions/technologies that are sustainable and environmentally friendly [1,2]. PGMs (Platinum Group Metals) play a crucial role in several of these technologies, including fuel cells, which are highly efficient and environmentally friendly energy sources, among other clean energy applications, such as water electrolysis and hydrogen storage [3,4]. PGMs, which include Pt (platinum), Pd (palladium), Rh (rhodium), Ru (ruthenium), Os (osmium), and Ir (iridium), have unique properties, such as high melting point, resistance to corrosion, and excellent catalytic activity, which make them indispensable in many fields of industrial applications, particularly in catalysis, due to their outstanding physical and chemical properties [5]. Some of these applications include the catalytic combustion of methane [6,7], recombination catalysts for safety devices [8], oxidation reactions for catalytic converters in automobiles [9], and medical devices [10].

Although Pt is the standard catalyst in many catalytic reactions, other PGMs and PGM alloys have also proven useful in this regard [11,12]. Alloying Pt with other PGMs and

non-PGMs has been considered [13–17]. For example, Pt/Pd alloys have been used in $H_2/O_2$ recombination reactions and have been found to provide enhanced catalytic activity, increased corrosion resistance, and improved mechanical strength [18]. Furthermore, Pt/Pd catalysts are used in numerous applications such as aromatic hydrogenation [7], methane combustion [6], NO oxidation over alumina support [19], and diesel oxidation catalysts (DOC) [20]. Due to the resistance of the alloy to poisons, catalyst deactivation, and wettability, Pt/Pd alloys are of particular interest as a potential recombination catalyst for nuclear safety [8,18,21–23]. Expanding the use of these catalysts for other applications requires a thorough understanding of their structural, mechanical, and thermodynamic properties.

Understanding the electronic, mechanical, and thermodynamic properties of alloys is crucial for the rational design of new catalysts. These properties provide a fundamental understanding of the stability, behaviour, and properties of materials under different conditions and enable accurate predictions of stable alloys. In manufacturing, the mechanical properties of materials, such as strength, ductility, hardness, toughness, elasticity, fatigue, and creep, are essential [24]. Of these, the elasticity of materials is particularly important, as it deals with the elastic stresses and strains, their relationship, and the external forces that cause them [25]. Therefore, predicting and controlling the mechanical properties of materials is crucial for creating structures that can withstand stresses and strains [26–28].

Although Pt/Pd alloys have been studied before [29], there are still gaps in understanding their thermodynamic properties. Therefore, the current study is necessary. Using density functional theory (DFT) calculations, the elastic, thermodynamic, electronic, and magnetic properties of Pt/Pd alloy structures were evaluated. The pristine Pt and Pd structures, as well as the Pt/Pd alloy structures, were found to be elastically and thermodynamically stable. It was observed that the percentage of Pt in the Pt/Pd alloy structures influenced the bulk modulus. As such, a higher bulk modulus and consequently increasing compression resistance, was observed in the Pt/Pd alloys, which consisted of higher Pt content. This paper is structured as follows: Section 2 presents the results and discussions, followed by a detailed description of the computational methodologies in Section 3. Finally, the conclusions are presented in Section 4.

## 2. Results and Discussion

### 2.1. Structural Stability

The initial pristine Pt and Pd structures, as well as the Pt/Pd alloy structures, are depicted using the convex hull in Figure 1. The $Pt_3Pd$ and $PtPd_3$ alloys are cubic structures with the 221 space group (Pm$\bar{3}$m). The $Pt_7Pd$ alloy structure is also cubic with the 225 space group (Fm$\bar{3}$m). However, the PtPd alloy structure is trigonal with the 166 space group (R$\bar{3}$m). The equilibrium volume data, along with the space groups and unit cell parameters, can be found in the supplementary information (Table S1).

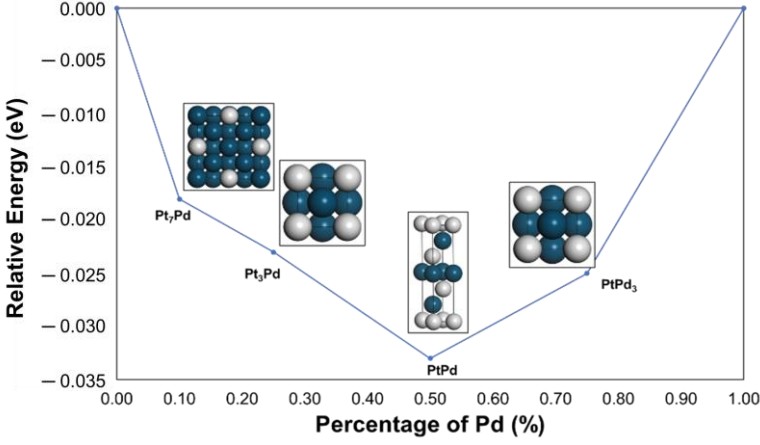

**Figure 1.** Relative energies of $PtPd_3$, PtPd, $Pt_3Pd$, and $Pt_7Pd$ alloy structures.

The respective equilibrium lattice constants for conventional cells, for the pristine Pt and Pd, as well as the alloy structures, as shown in Figure 2, were obtained by fitting the total energy versus volume to the Murnaghan equation of state [30]. Figure 2 illustrates the equilibrium volumes obtained for the pristine Pt and Pd structures, as well as the Pt/Pd alloy structures. The equilibrium volumes are as follows: 59 Å$^3$ for pristine Pt and Pd, 59–60 Å$^3$ for PtPd$_3$ and Pt$_3$Pd, 89 Å$^3$ for PtPd, and 483 Å$^3$ for Pt$_7$Pd. The equilibrium volumes are consistent with the space groups and crystal structures considered as illustrated in the supplementary information (Figure S1). The equilibrium lattice constants were 3.93 Å (Pt), 3.89 Å (Pd), 3.90 Å (PtPd$_3$), 2.771 Å (PtPd), 3.92 Å (PdPt$_3$), and 7.849 Å (Pt$_7$Pd), respectively. As shown in Table 1, the calculated lattice constants were consistent with those obtained in previous studies [31].

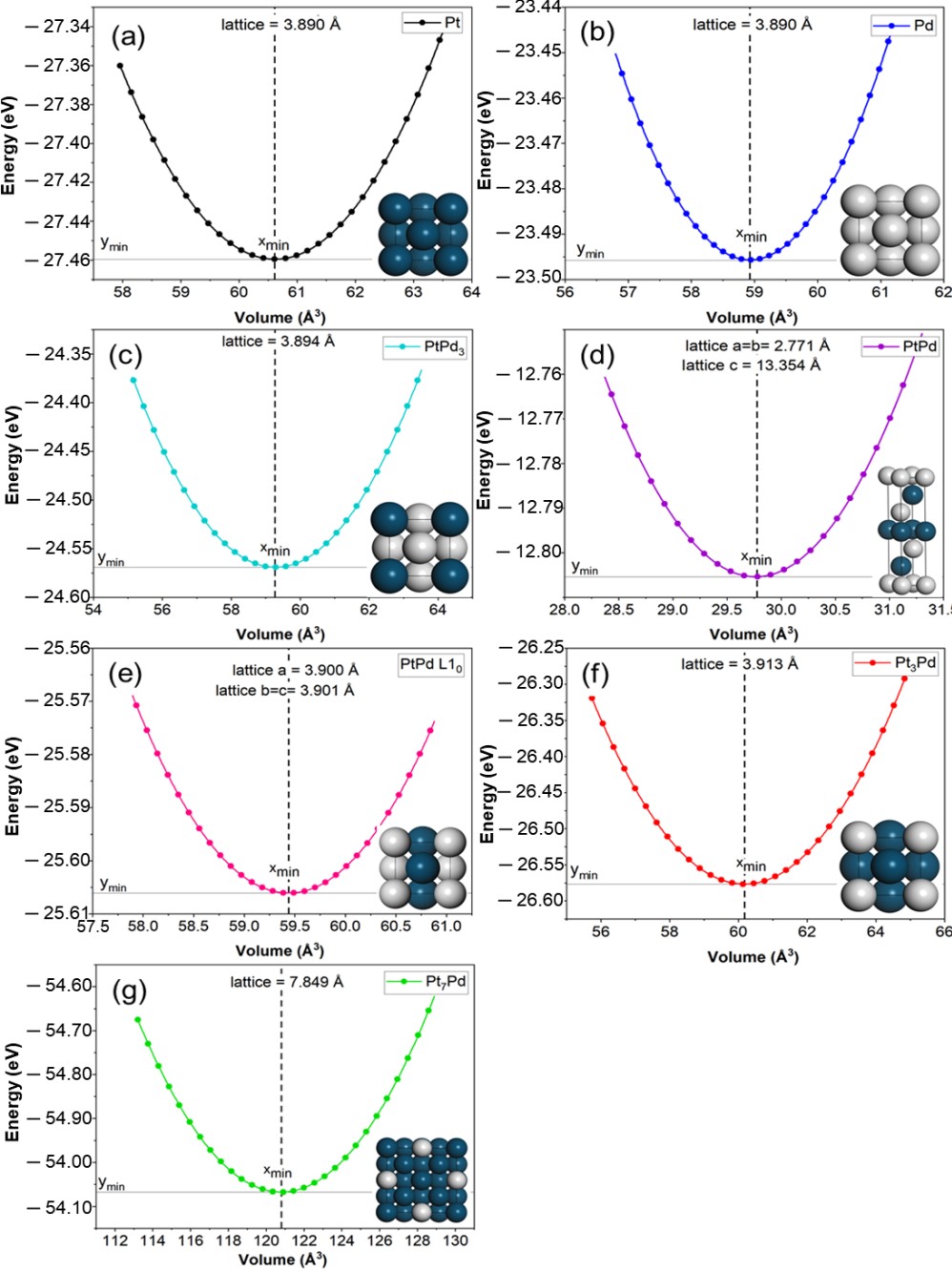

**Figure 2.** Murnaghan's fit for the pristine (**a**) Pt and (**b**) Pd, as well as (**c**) PtPd$_3$, (**d**) PtPd, (**e**) PtPd (L1$_0$), (**f**) Pt$_3$Pd, and (**g**) Pt$_7$Pd alloy structures (conventional unit cell structures).

Previous experimental studies reported the existence of a PtPd alloy configuration [32–34]. In 1959, Raub et al., reported that the PtPd alloy can exist as a cubic structure with lattice constants of 3.90 Å, an observation that was also made by Darby et al. in 1972, who also reported that the cubic structure with lattice constant of 3.903 Å. In 2011, a computational study by Zhiyao et al. [35] reported that the cubic structure of PtPd belongs to the $L1_0$ phase with computed lattice constants of a = b = 3.964 Å and c = 3.966 Å. In the current study, the stable alloy configurations were predicted using the convex hull approach, which showed the existence of stable alloy configuration of PtPd in the hexagonal rhombohedral phase, characterised by the R3M space group with lattice constants of a = b = 2.771 Å and c = 13.354 Å.

Taking into account previous studies, the present study further investigated the possibility of the reported experimental structures. Using the structure observed by Raub et al. [33], the equilibrium lattice constant calculated using DFT is found to be a = 3.900 Å and b = c = 3.901 Å, which is consistent with the $L1_0$ phase reported by Zhiyao et al. [35] Upon the calculation of the formation energies, the formation energy of the $L1_0$ phase was −0.128 eV while that of the predicted hexagonal rhombohedral was −0.050 eV, indicating ease of formation of the latter rather than the former. Phonon calculations also indicated that this configuration was stable in both the $L1_0$ and hexagonal rhombohedral phases.

The lattice constants of alloys obey either Vegard's law or Retger's law, but cannot simultaneously satisfy both laws [36]. Vegard's law states that the lattice parameters of a continuous substitutional solid solution vary linearly with concentration at constant temperature when the bonding nature is similar in the constituent phases. On the other hand, Retger's law states that the mole fraction and cell volume of an alloy have a linear variation with each other. Consequently, alloys with higher mole fractions exhibit higher cell volumes [36].

In this study, the lattice constants of the $Pt_3Pd$ and $PtPd_3$ alloy structures were found to deviate linearly from those of pristine Pt and Pd, thus adhering to Vegard's law. However, it is important to note that Vegard's law is an approximation, and binary alloys can deviate significantly from it in certain cases [37–41]. The $Pt_3Pd$ and $PtPd_3$ alloy structures exhibit mole fractions and cell volume (14 atoms and volumes of 59–60 Å$^3$) similar to the pristine Pt and Pd structures.

In contrast, the $Pt_7Pd$ and PtPd alloy structures, with different mole fractions and cell volumes, in contrast to the pristine Pt and Pd structures, follow Retger's law. The $Pt_7Pd$ alloy has a mole fraction of 63 atoms and a cell volume of 483 Å$^3$, while the PtPd alloy has a mole fraction of 16 atoms and a cell volume of 89 Å$^3$.

Therefore, the differences in lattice constants and volumes observed in the Pt/Pd alloy structures are a consequence of the varying mole fraction. This indicates that the configuration of Pt/Pd alloys can be properly described by the convex hull method, taking into account the composition-dependent changes in lattice parameters and volume.

The formation energy ($E_f$) of the respective pristine Pt and Pd, as well as the Pt/Pd alloy structures, was calculated using Equation (1) [42].

$$E_f^{PtPd} = E_{tot}^{PtPd} - N \times E_{Pt}^{bulk} - N \times E_{Pd}^{bulk} \tag{1}$$

where $E_{tot}^{PtPd}$ denotes the total energy of Pt/Pd alloy structures; $E_{Pt}^{bulk}$ and $E_{Pd}^{bulk}$ denote the total energies of bulk Pt and Pd, respectively; and N denotes the number of atoms of each species (Pt and Pd atoms) in the alloy structure. Formation energy determines the change in energy when an alloy structure is created from its constituent elements. Thus, the negative formation energies imply that the Pt/Pd alloy structures are energetically stable and likely to form spontaneously. The calculated formation energies for the alloy structures are also consistent with previous studies [43], as shown in Table 1. All calculated formation energies as shown in Table 1 were negative, indicating that the formation of Pt/Pd alloy structures from their constituent elements is favourable. As such, the Pt/Pd alloy structures are energetically stable. Note that the PtPd in the hexagonal rhombohedral phase and

L1$_0$ phase are both energetically stable, with the latter being energetically more stable. However, both phases of PtPd have been used for further evaluation. In the next section, the elastic, phonon, and thermodynamic stability of these alloy structures are presented.

**Table 1.** Calculated equilibrium lattice constants for the conventional cells, in angstroms (Å), Volume in cubic angstroms (Å$^3$), space groups, and formation energies of pristine Pt and Pd, as well as the Pt/Pd alloy structures in electron volt per formula unit (f.u).

| | Unit Cell Parameters (Å) | | | Unit Cell Angles (°) | | | Equilibrium Volume (Å$^3$) | Space Group (no) | Formation Energy (eV/f.u) |
|---|---|---|---|---|---|---|---|---|---|
| | **a** | **b** | **c** | **α** | **β** | **γ** | | | |
| Pd | 3.890 | 3.89 | 3.890 | 90 | 90 | 90 | 58.86 | 225 | |
| Pd ref | 3.89 [a,b] | 3.890 | 3.89 | | | | | | |
| PtPd$_3$ | 3.890 | 3.890 | 3.890 | 90 | 90 | 90 | 58.86 | 221 | −0.023 |
| PtPd$_3$ ref | 3.88 [c] | 3.88 | 3.88 | 90 | 90 | 90 | | | |
| PtPd | 2.771 | 2.771 | 13.354 | 90 | 90 | 120 | 88.80 | 166 | −0.050 |
| PtPd (L1$_0$) | 3.900 [d] | 3.901 [d] | 3.901 [d] | 90 [d] | 90 [d] | 90 [d] | 59.36 | 123 | −0.128 |
| Pt$_3$Pd | 3.920 | 3.920 | 3.920 | 90 | 90 | 90 | 60.24 | 221 | −0.025 |
| Pt$_3$Pd ref | 3.91[c] | 3.91 | 3.91 | 90 | 90 | 90 | | | |
| Pt$_7$Pd | 7.789 | 7.789 | 7.789 | 90 | 90 | 90 | 483.53 | 225 | −0.018 |
| Pt | 3.930 | 3.930 | 3.930 | 90 | 90 | 90 | 60.70 | 225 | |
| Pt ref | 3.92 [b,c] | 3.92 | 3.92 | | | | | | |

Pt$^{ref}$, Pd$^{ref}$ etc. refers to referenced papers. [a] Ref. [29], [b] Ref. [32], [c] Ref. [43], [d] Refs. [32,33,35].

### 2.2. Elastic, Phonon, and Thermodynamic Properties

The detailed mathematical equations and a comprehensive description of how to determine the bulk modulus (B), shear modulus (G), Young modulus (E), and Poisson's ratio (*v*) from the elastic constants (C$_{ij}$) can be found in the supplementary section.

### 2.2.1. Elastic Properties

Mechanical stability can be explored through the determination of elastic constants. The symmetry of the structure determines the elastic constants (C$_{ij}$) needed to evaluate the mechanical properties. The elastic stability (at zero pressure) of the pristine Pt and Pd structures, as well as the alloys considered, was determined using the stress-strain method. The computed elastic constants are given in Table 2. The born stability criteria [44] for cubic (2) and trigonal (3) systems are as follows:

$$C_{11} - C_{12} > 0; \quad C_{11} + 2C_{12} > 0; \quad C_{44} > 0 \tag{2}$$

$$C_{11} - C_{12} > 0; \quad C_{13}{}^2 < 0.5 \times C_{33}(C_{11} + C_{12}); \quad C_{14}{}^2 < 0.5 \times C_{44}(C_{11} - C_{12}); \quad C_{44} > 0 \tag{3}$$

Based on the cubic symmetry of the Pt, Pd, Pt$_3$Pd, PtPd$_3$, and Pt$_7$Pd structures, three independent elastic constants (C$_{11}$, C$_{12}$, and C$_{44}$) are required to analyse the mechanical behaviour of these alloy structures. Furthermore, the alloy structure of PtPd, which is trigonal, requires six independent elastic constants (C$_{11}$, C$_{12}$, C$_{14}$, C$_{33,}$ and C$_{44}$). The considered structures satisfied Born's stability criteria as presented in Equations (2) and (3). Therefore, they are elastically stable.

Several elastic constants are related to the modulus of elasticity, depending on the applied strain direction, such as lateral, axial, torsional, and compression [24]. The bulk modulus of pristine Pt and Pd, and the considered Pt/Pd alloy structures determine their compression resistance, which is calculated using Murnaghan's equation of state [30]. The values of the bulk modulus (B) and the shear modulus (G) represent the extreme limits of the elastic moduli [45] and are expressed differently depending on the crystal

symmetry [46]. B is a measure of the resistance of a material to compression, while G is related to the applied shear strain or rigidity [47]. In Table 2, B and G are calculated from the elastic constants are shown. The calculated B for the pristine Pt and Pd, as well as the considered Pt/Pd alloy structures, are consistent with published values [31,48,49]. As shown in Table 2, the value of B increases with increasing Pt content in the alloy. The high values of B and G indicate that the pristine Pt and Pd, as well as the considered Pt/Pd alloys, are resistant to compression and shear deformation.

The Young modulus (E) is an important mechanical property of materials that refers to the behaviour of the material towards bending/twisting deformation (stiffness), whereas the Poisson's ratio (*v*) relates to the absolute ratio of the lateral strain (stretching and compression) in response to longitudinal strain [24]. Therefore, stiffer materials are thus associated with high E values in the range of 150–550 GPa [24]. The E values, presented in Table 2, indicate that the pristine Pt and Pd, as well as the Pt/Pd alloys considered, are all high and therefore will resist deformation at low-stress loads. The calculated *v* ratios obtained for the pristine Pt and Pd, and the considered Pt/Pd alloy structures are in the range of 0.30–0.44 GPa [50,51], indicating that these considered structures are prone to some measure of lateral deformation when under compression. A *v* ratio of < 0.33 GPa implies that the system is metallic [52]. Therefore, from the ratios presented in Table 2, the pristine Pt and Pd, as well as the Pt/Pd alloy structures considered are metallic. Therefore, it is evident that the considered Pt/Pd alloy structures have a high resistance to compression and deformation while also exhibiting elasticity.

**Table 2.** The elastic constants ($C_{ij}$), bulk modulus (B), shear modulus (G), Young modulus (E), and Poisson (*v*) for pristine Pt and Pd, as well as the Pt/Pd alloy structures. The table shows the minimum and maximum values for G, E and *v*.

| Phase | $C_{11}$ GPa | $C_{12}$ GPa | $C_{13}$ GPa | $C_{14}$ GPa | $C_{33}$ GPa | $C_{44}$ GPa | B GPa | G GPa | E GPa | N GPa |
|---|---|---|---|---|---|---|---|---|---|---|
| Pt | 327.76 | 253.50 | | | | 82.19 | 278 | 46–82 | 107–224 | 0.37–0.44 |
| Pt [ref] | | | | | | | 299 [a], 238 [b] 250 [c], 262 [d] | 46–54 [e] | 132–164 [e] | 0.48–0.51 [e] |
| Pd Pd [ref] | 232.90 | 173.56 | | | | 91.06 | 193 221 [a] 172 [b] 181 [d] | 38–91 | 85–236 | 0.30–0.43 |
| PtPd₃ | 256.55 | 183.97 | | | | 86.67 | 216 | 47–90 | 107–237 | 0.32–0.42 |
| PtPd | 335.21 | 190.62 | 191.14 | 6.58 | 323.87 | 72.30 | 227–249 253 [a] | 70–78 | 189–211 | 0.36–0.37 |
| Pt₃Pd | 327.20 | 215.01 | | | | 91.72 | 217 | 45–72 | 108–193 | 0.35–0.42 |
| Pt₇Pd | 372.38 | 210.00 | | | | 73.76 | 264 | 74–81 | 202–221 | 0.36–0.37 |

Pt[ref], Pd[ref] etc refers to referenced papers. [a] Ref. [29], [b] Ref. [48], [c] Ref. [49], [d] Ref. [43], [e] Ref. [5].

## 2.2.2. Phonon Properties

The calculated phonon band structures and DOS plotted along the high symmetry points of the respective Brillouin zones are shown in Figures 3–7. No imaginary modes (negative frequencies) were observed in the Pt/Pd alloy structures. Therefore, the Pm3m (Pt₃Pd, PtPd₃), Fm3m (Pt₇Pd), and R3m (PtPd) phases of the considered Pt/Pd alloy structures are dynamically stable.

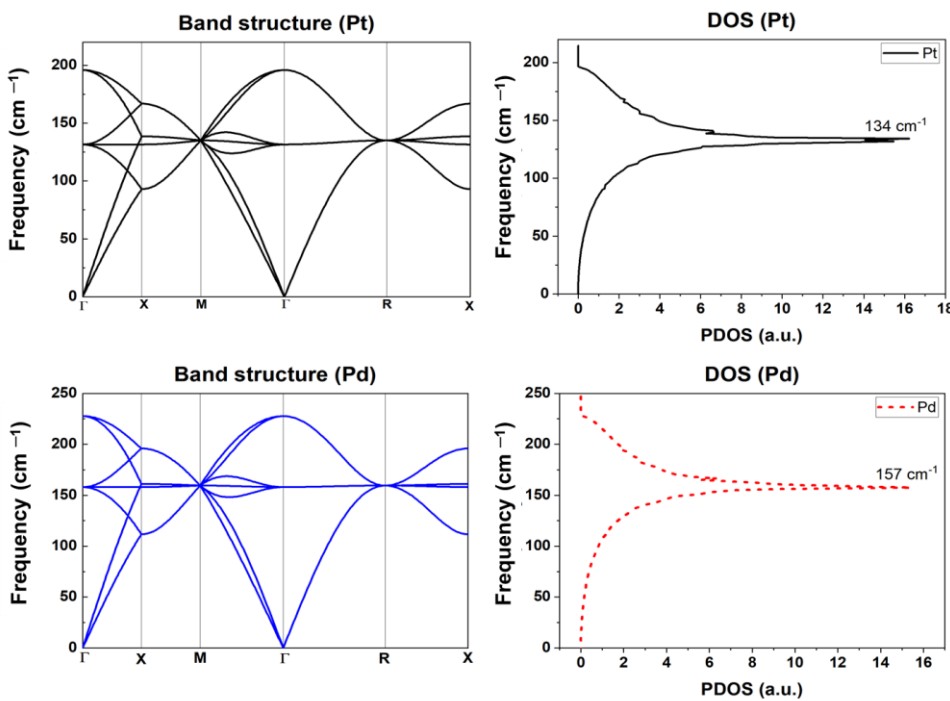

**Figure 3.** Comparison between calculated phonon band spectra and phonon DOS spectra for the pristine Pt and Pd structures.

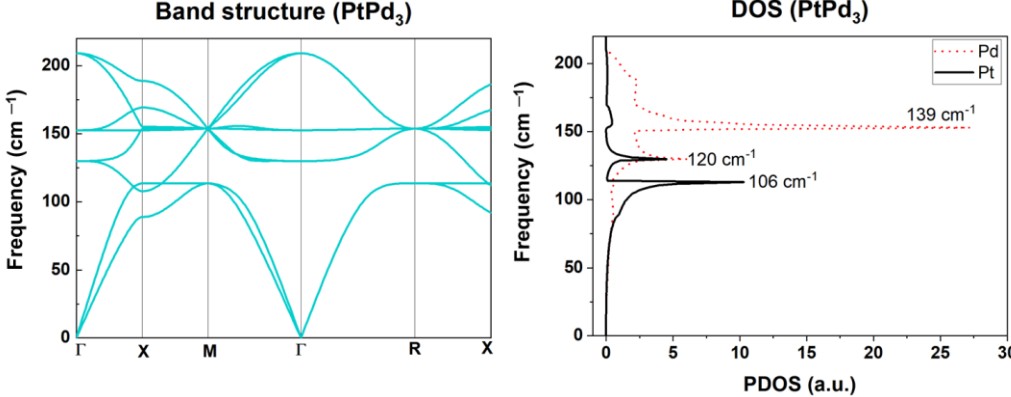

**Figure 4.** Comparison between the calculated phonon band spectra and phonon DOS spectra for the considered PtPd$_3$ alloy structure.

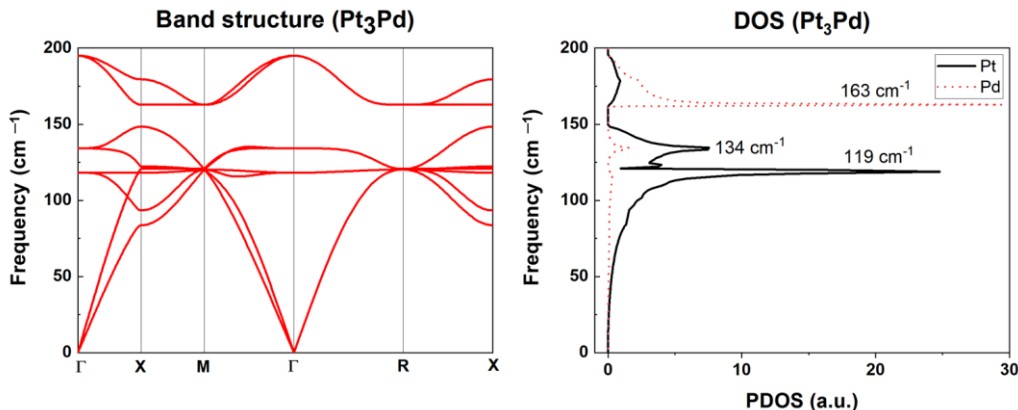

**Figure 5.** Comparison between the calculated phonon band spectra and phonon DOS spectra for the considered Pt$_3$Pd alloy structure.

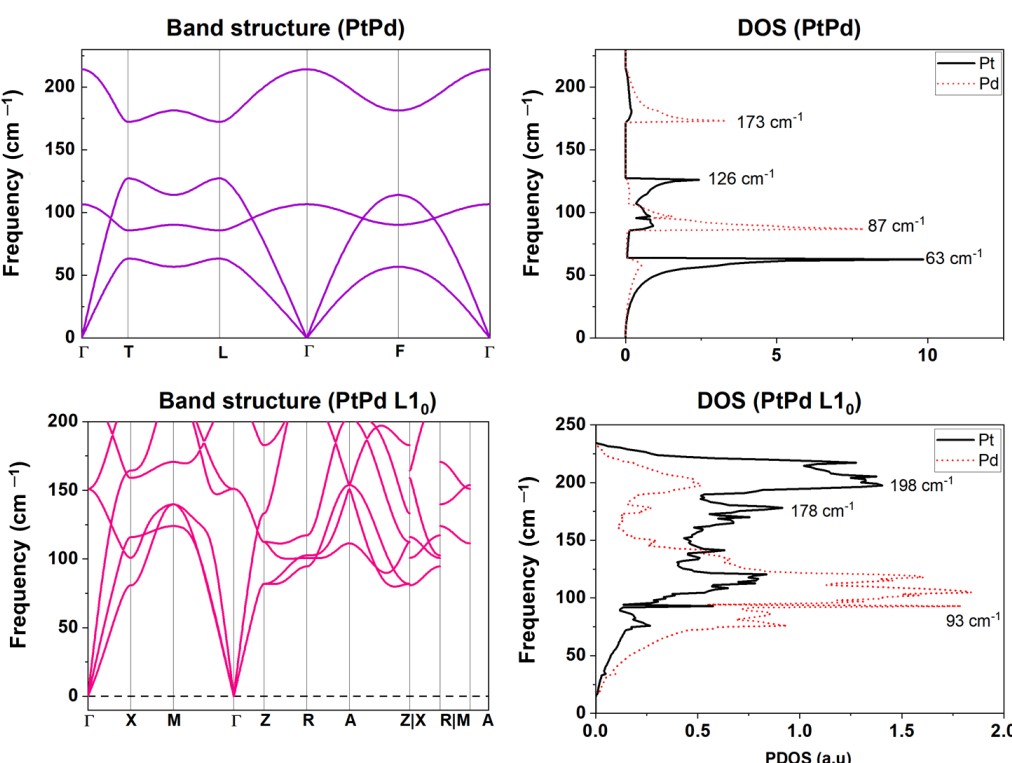

**Figure 6.** Comparison between the calculated phonon band spectra and phonon DOS spectra for the PtPd structures in the hexagonal rhombohedral and L1$_0$ phases.

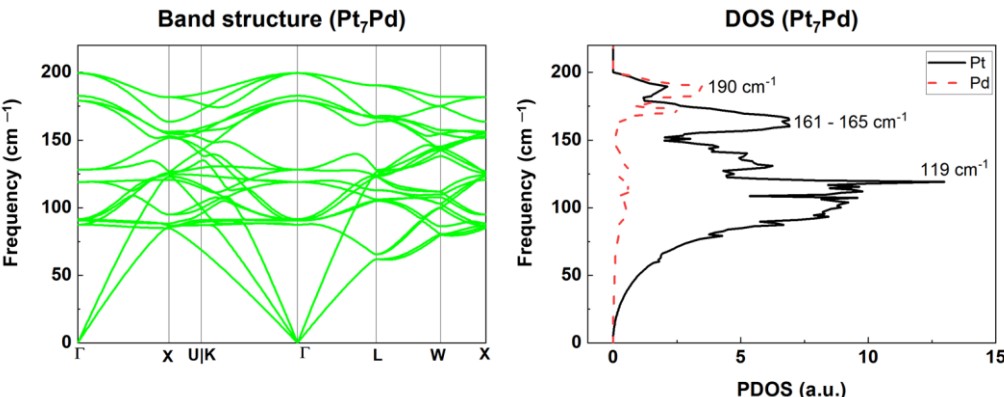

**Figure 7.** Comparison between the calculated phonon band spectra and phonon DOS spectra for the considered Pt$_7$Pd alloy structure.

The dynamic stability of the alloys was further assessed by examining their pair distribution function (PDF) for a determined temperature range [53]. PDF is used as a descriptor of dynamic stability since it describes the short-range order between atoms in different materials [54]. In this study, molecular dynamics (MD) simulations were conducted to determine the PDF for both pristine Pt and Pd structures, as well as for Pt/Pd alloy structures, within a temperature range of 100 K to 2100 K. For dynamic stability structures, a decrease in temperature causes the pair correlation function to become more structured, reflecting the ordering of atoms in the crystalline lattice. For detailed information and results, refer to Figure S3 in the Supplementary Information. These results indicate a high degree of dynamical stability in the pristine Pt and Pd, as well as the Pt/Pd alloy structures, exhibit a high degree of dynamic stability and maintain their integrity as the temperature increases.

The phonon DOS for the pristine Pt and Pd, as shown in Figure 3, indicates peaks of the frequency spectrum at 134 cm$^{-1}$ and 157 cm$^{-1}$ respectively.

Figures 4 and 5 show the calculated phonon band structures and DOS for the Pt$_3$Pd, PtPd, and PtPd$_3$ alloy structures. The DOS indicates that the Pt atom is dominant at lower frequencies (between the ranges of 113–134 cm$^{-1}$) for the PtPd$_3$ (Figure 4) and Pt$_3$Pd (Figure 5) alloy structures, whereas the Pd atom is dominant at higher frequencies with peaks at 153 cm$^{-1}$ for PtPd$_3$, and 163 cm$^{-1}$ for Pt$_3$Pd alloys. Additional peaks were observed at 113 cm$^{-1}$ (PtPd$_3$), 119 cm$^{-1}$ (Pt$_3$Pd), 130 cm$^{-1}$ (PtPd$_3$), and 134 cm$^{-1}$ (Pt$_3$Pd).

Figure 6 shows the PtPd structures with an equal number of Pd and Pt atoms. In the hexagonal rhombohedral phase, the Pt and Pd atoms were observed to be dominant at lower frequencies (30–66 cm$^{-1}$) and higher frequencies (84–110 cm$^{-1}$ and 171–200 cm$^{-1}$), respectively. Furthermore, several other peaks for the Pt and Pd atoms were observed in the DOS profile at 63 cm$^{-1}$ (Pt), 87 cm$^{-1}$ (Pd), 126 cm$^{-1}$ (Pt), and 173 cm$^{-1}$ (Pd). On the other hand, in the L1$_0$ phase the Pd atoms were observed to be dominant at lower frequencies (76–121 cm$^{-1}$) and Pt at higher frequencies (141–217 cm$^{-1}$).

In Figure 7, the dominant frequency contribution is from the Pt atoms. This is due to the higher Pt content in the alloy, while the Pd frequency peak is observed around 176 cm$^{-1}$.

Overall, the Pt and Pd frequency modes play a role in the dynamical stability of the considered Pt/Pd alloys. This observation is consistent with previous studies in which the mass of the elements determines the position of the phonon frequencies [55,56].

### 2.2.3. Thermodynamic Properties

A detailed mathematical equation and a comprehensive description of the thermodynamic properties are available in the Supplementary Information. The impact of temperature on alloy properties, such as crystal structure and thermal properties, is important [57–59]. Thus, the thermal properties were calculated in this study from the phonons. The relationship between temperature and Helmholtz free energy ($F$), entropy ($S$), and constant-volume specific heat ($C_v$) is shown in Figure 8. The relationship between these properties is given in Equations (4)–(6), respectively, for Helmholtz free energy (4), entropy (4) and constant-volume specific heat (5) [60]. In these equations, the Helmholtz free energy is related to entropy and temperature (4), whereas constant-volume specific heat is expressed in terms of the second derivative of Helmholtz free energy with respect to temperature at a constant volume (6) [60].

$$F = U - TS \qquad (4)$$

$$C_v = \left(\frac{dU}{dT}\right)v \qquad (5)$$

$$C_v = -T\left(\frac{d^2F}{dT^2}\right) \qquad (6)$$

Entropy is an important thermodynamic property that affects a metal's heat transfer and energy exchange. In Figure 8a, the calculated entropies for pristine Pt and Pt$_3$Pd alloys are approximately similar in the considered temperature range, which indicates similar ordering of the structures. A similar observation was made in the case of pristine Pd and PtPd$_3$ alloy structures. In comparison to the pristine Pt and Pd structures, the entropy of the Pt$_7$Pd alloy increases rapidly at a temperature range of 0 to 300 K, while it increases slowly in the case of PtPd alloy in the same temperature range. The overall entropy for the considered structures is the highest for the Pt$_7$Pd alloy structure and lowest for the PtPd alloy structure. Low entropy values indicate ordered structures, which allows for better thermal conductivity and reduces the likelihood of thermal expansion and deformation. Therefore, in comparison with the pristine Pt and Pd structures, the PtPd alloy will possess better thermal conductivity and reduced thermal expansion.

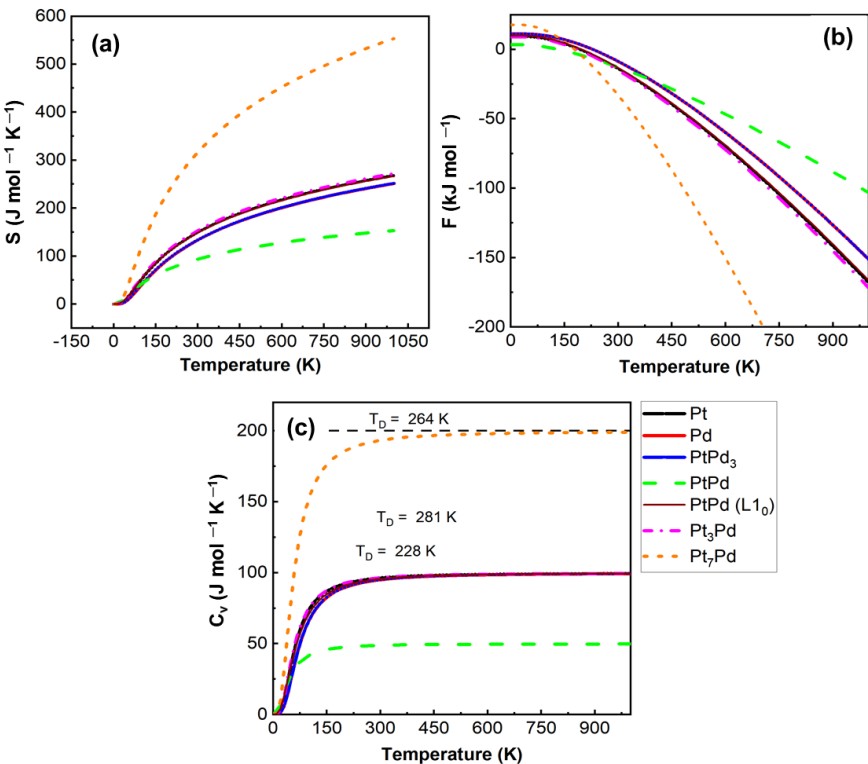

**Figure 8.** Calculated temperature dependence of the (**a**) entropy (S), (**b**) Helmholtz free energy (F), and (**c**) constant-volume specific heat capacity ($C_v$).

In Figure 8b, the calculated Helmholtz free energy shows an inverse relationship with the entropy, as shown in Figure 8a. The calculated constant-volume specific heat ($C_v$) increases exponentially as a function of temperature up to ~250 K and then flattens out (see Figure 8c) as it approaches the Dulong–Petit asymptote. The $C_v$ at this asymptote is 200 J/mol K for $Pt_7Pd$; 100 J/mol K for $Pt_3Pd$, $PtPd_3$, Pt; and Pd, and 50 J/mol K for PtPd. The Debye temperatures calculated from elastic constants were 301 K (Pd), 281 ($PtPd_3$), 280 K (PtPd), 270 K ($Pt_3Pd$), 264 K ($Pt_7Pd$) and 228 K (Pt). We found that the Debye temperature decreased with increasing Pt content, similar to the observation of Tang et al. [43].

Overall, the pristine Pt and Pd structures, as well as the Pt/Pd alloy structures, exhibited similar trends for the investigated thermodynamic properties. Specifically, an increase in temperature resulted in an increase in entropy, a decrease in free energy, and an increase in the specific heat capacity for all structures considered. Although the trends are the same for all structures investigated, the thermodynamic properties vary. The entropy, Helmholtz free energy, and specific heat capacity of the $Pt_3Pd$ alloy structure is similar to those of the pristine Pt structure, whereas those of the $PtPd_3$ alloy structure are similar to those of the pristine Pd structure.

### 2.3. Electronic Properties

The projected density of states (PDOS) for the *d*-electrons of the pristine Pt and Pd metals, as well as the Pt/Pd alloy structures, is shown in Figure 9. The calculated electronic structures indicate that the pristine Pt and Pd, as well as the Pt/Pd alloy structures, are metallic due to the absence of an electronic band gap at the Fermi level ($E_F = 0$). There is an equal number of electrons in the up (majority) and down (minority) spin channels within the full *d* orbitals. As such the pristine Pt is non-magnetic with a symmetric PDOS, which is in agreement with previous work [61]. For the pristine Pd structure, a slight asymmetry is observed in the PDOS close to the Fermi level, which indicates a paramagnetic ground state.

The atomic spin moment ($\mu_B$ atom$^{-1}$) obtained from Bader analysis for the pristine Pd is 0.28. The Pt/Pd alloy structures considered in this study are non-magnetic, like pristine Pt.

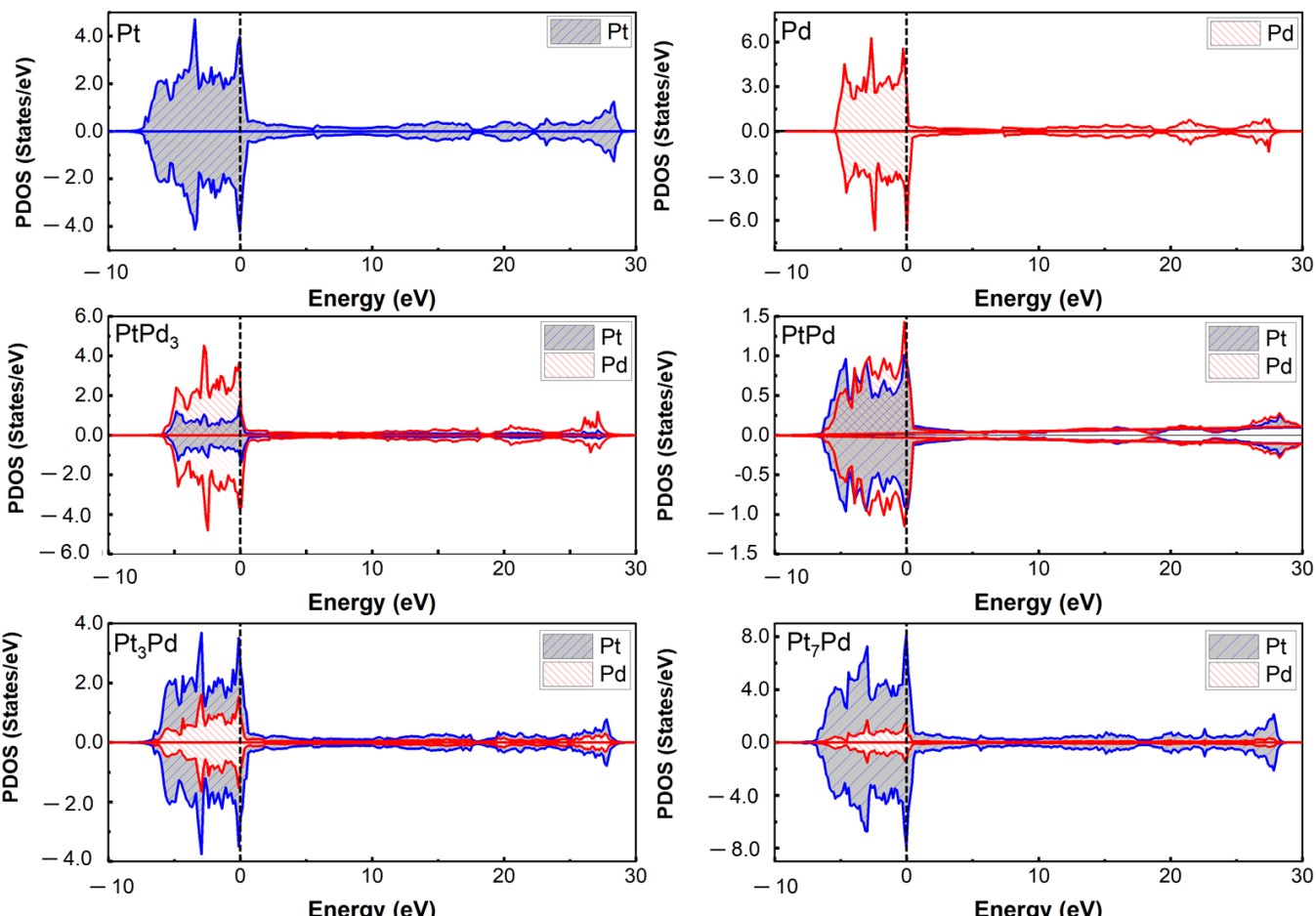

**Figure 9.** The calculated projected density of states (PDOS) for each atom in the pristine Pt and Pd, as well as the considered Pt/Pd alloy structures.

The calculated atomic charges indicate that the Pt and Pd atoms were found to be charge-neutral, as shown in Table 3. In the case of the considered Pt/Pd alloys, the Pd atoms were negatively charged, while the Pt atoms were positively charged, resulting in a net charge of zero.

**Table 3.** Atomic charge (q) per atom for the pristine Pt and Pd, as well as for the Pt/Pd alloy structures.

| Structure | Atom | q (e/atom) |
|:---:|:---:|:---:|
| Pt | Pt | 0.00 |
| Pd | Pd | 0.00 |
| PtPd$_3$ | Pt | −0.10 |
| | Pd | 0.03 |
| PtPd | Pt | −0.11 |
| | Pd | 0.11 |
| Pt$_3$Pd | Pt | −0.03 |
| | Pd | 0.09 |
| Pt$_7$Pd | Pt | −0.03 |
| | Pd | 0.24 |

### 3. Computational Method

The pristine Pt and Pd as well as Pt/Pd alloy structures were investigated using spin-polarised density functional theory calculations (DFT) as implemented in the Vienna Ab Initio Simulation Package (VASP) code [62–64]. The generalised-gradient approximation (GGA) functional developed by Perdew, Burke, and Ernzernhof (PBE) [65] was applied. The interaction between core and valence electrons for Pd and Pt was described using the projected augmented-wave (PAW) method [66,67].

The structural optimisation was carried out using a kinetic energy cut-off fixed at 500 eV for the plane-wave basis set expansion of the Kohn–Sham (KS) valence states with an electronic convergence set at $10^{-6}$ eV. The Methfessel−Paxton approach was used with a smearing width of 0.2 eV. The alloy configurations ($Pt_3Pd$, $PtPd_3$, Pt7Pd, and PtPd) were obtained from the open quantum materials database (OQMD) [68]. The alloy structures of $Pt_3Pd$, PtPd, $PtPd_3$, and $Pt_7Pd$ with FCC and trigonal crystal structures were used, while the structure of the $L1_0$ phase of the alloy configuration of PtPd was obtained from the experimental work of Raub et al. [33] The Brillouin zones were sampled using $\Gamma$-centred Monkhorst-Pack grid [69] with a spacing width of 0.03 Å. Phonon frequencies were evaluated using PHONOPY [70]. A finite displacement amplitude of 0.01 Å was used, along with a Monkhorst sampling width of 0.04 Å for the phonon density of state calculations. The Monkhorst sampling width was set to 0.02 Å for the density of states (DOS) calculations, along with Blöchl corrections.

In this study, the equilibrium properties were estimated for each of the alloy configurations using Murnaghan's equation of state. The bulk modulus, shear modulus, Young modulus, Poisson's ratio, and elastic constants (stress-strain method) of these alloys were evaluated as implemented in the VASPKIT module [71]. Additionally, the thermodynamic properties were extracted from phonon calculations. For repeatability, the atomic coordinates and computational parameters utilised in this study have been included in the supplementary information.

### 4. Conclusions

This study has provided insights into the structural, elastic, mechanical, and thermal properties of pristine Pt and Pd, as well as various Pt/Pd alloy structures. The study found that pristine Pt and Pd, as well as the considered $PtPd_3$, PtPd, $Pt_3Pd$, and $Pt_7Pd$ alloy structures, are energetically stable and mechanically stable. The $PtPd_3$, PtPd, $Pt_3Pd$, and $Pt_7Pd$ alloy structures are energetically stable due to their negative formation energies of $-0.023$ eV, $-0.033$ eV, $-0.025$ eV, and $-0.018$ eV, respectively. Furthermore, the Pt/Pd alloy structures are mechanically stable, as they satisfy Born's stability criteria. The phonon calculations indicate that the considered structures are dynamically stable due to the absence of imaginary modes. Moreover, the phonon DOS for the considered Pt/Pd alloy structures indicated that the Pt atom is the dominant contributor at low-frequency modes in the range of 80–133 cm$^{-1}$, while the Pd atom is the dominant contributor at high-frequency modes at frequencies above 140 cm$^{-1}$. The entropy, Helmholtz free energy, and constant-volume specific heat capacities were found to be similar to the considered temperature for both the pristine Pt and $Pt_3Pd$ alloy structures, as well as for the Pd and $PtPd_3$ alloy structures. Additionally, the Debye temperature was found to decrease with increasing Pt content, hence the considered Pt/Pd alloy structures were found to be ordered thermodynamically stable structures. The considered Pt/Pd alloy structures were also found to exhibit metallic characteristics and to be non-magnetic. Overall, these findings contribute to a better understanding of the properties and stability of Pt/Pd alloy structures, which are relevant in various fields, including materials science and catalysis.

**Supplementary Materials:** The following supporting information can be downloaded at: https://www.mdpi.com/article/10.3390/condmat8030076/s1, Figure S1: Volume calculation for cubic Pt/Pd alloy structures, where $\alpha = \beta = \gamma = 90°$. Figure S2: Volume calculation for non-cubic Pt/Pd alloy structure where $\alpha = \beta = \gamma \neq 90°$. Figure S3: Temperature behaviour of the pair distribution function for the Pt-Pd bond distances, obtained from molecular dynamics (MD) simulations conducted over the range of 100 K to 2100 K. The plots represent the pristine structures of (a) Pt, (b) Pd, as well as the alloy structures for (c) PtPd3, (d) PtPd, (e) Pt3Pd and (f) Pt7Pd. Table S1: Atomic coordinates of the Pt/Pd alloy structures, as implemented in VASP. Table S2: INCAR file for Geometric optimisation of bulks. Table S3: INCAR file for density of States and work function calculations. Table S4: INCAR file for determining elastic constants. Table S5: Input file (VPKIT.in) for determining elastic constants. Table S6: CONTCAR file for Phonopy displacement 001 (PtPd3). Table S7: CONTCAR file for Phonopy displacement 002 (PtPd3). Table S8: INCAR file for Phonopy calculation of sets of forces. Table S9: mesh.conf file for Phonopy optimisation. Table S10: pdos.conf file for Phonopy optimisation. Table S11: Band.conf file for Phonopy. Table S12: INCAR file for MD simulations with changing temperature. Table S13: KPOINT file for MD simulations. The following citations was used in the supplementary section, Mathematical equations for the description of elastic properties [72–74] and temperature properties [75,76].

**Author Contributions:** L.M.B.: Investigation, Validation, Data curation, Visualization, Writing—original draft. C.N.M.O.: Writing—review & editing. K.O.O.: Writing—review & editing. D.G.B.: Conceptualisation, Resources, Writing—review & editing, Supervision, Project administration, Funding acquisition. D.L.S.: Conceptualisation and proofing. P.S.V.: Conceptualisation and proofing. E.I.P.: Conceptualisation and proofing. All authors have read and agreed to the published version of the manuscript.

**Funding:** This research was funded by the Department of Science and Innovation (DSI) and the HySA Infrastructure Center of Competence at the North-West University (NWU), South Africa; and Norilsk Nickel Asia Ltd. (Hong Kong); are acknowledged for financial support through the KP5 program.

**Data Availability Statement:** The data presented in this study are available in the supplementary material.

**Acknowledgments:** The computational resources were provided by South Africa's Centre for High-Performance Computing (CHPC).

**Conflicts of Interest:** The authors declare no conflict of interest.

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
