# Peer review of "Ab Initio Study of Structural, Electronic, and Thermal Properties of Pt/Pd-Based Alloys"

_condensedmatter, doi:10.3390/condmat8030076_

Round 1
Reviewer 1 Report
In this work, the authors studied the structural, elastic, mechanical, and thermal properties structural, electronic, and thermal properties of Pt/Pd based alloys. The result can deepen the understanding of the properties and stability of Pt/Pd alloy structures. The topic is interesting. However, there are still some comments and suggestions for the authors consideration.
1. The percentage of Pt should influence the bulk properties. The reason why these alloys of Pt3Pd, PtPd3, Pt7Pd, and PtPd were selected is poorly presented. It will be interesting to know how the authors will argue the selection of these alloys, since there does not contain any experiments.
2. There are some “Error! Reference source not found” in the manuscript. The authors need to carefully check the cite of refence.
3. The author should give Pt/Pd alloy configurations as insets in figure 1, which can help readers to understand the Pt/Pd alloy structures.
4. In Eq.(1), the definition of the formation energy is obscure. Since the Pt/Pd bulks have different metal atoms, the formation energy would be much better to be defined with the unit of eV/atom as that in ref. 34.
Minor editing of English language required.
Author Response
- The percentage of Pt should influence the bulk properties. The reason why these alloys of Pt3Pd, PtPd3, Pt7Pd, and PtPd were selected is poorly presented. It will be interesting to know how the authors will argue the selection of these alloys, since there does not contain any experiments.
Response: This issue has been addressed in the manuscript, Figure 1.
Statement in the manuscript:
“The initial pristine Pt and Pd structures, as well as the Pt/Pd alloy structures, are depicted using the convex hull in Figure 1. The Pt3Pd and PtPd3 alloys are cubic structures and belong to the 221 space group (Pmm). The Pt7Pd alloy structure, on the other hand, is also cubic and belongs to the 225 space group (Fmm). On the other hand, the PtPd alloy structure is trigonal and belongs to the 166 space group (Rm ).“
- There are some “Error! Reference source not found” in the manuscript. The authors need to carefully check the cite of refence.
Response: This issue has been addressed throughout the manuscript.
- The author should give Pt/Pd alloy configurations as insets in figure 1, which can help readers to understand the Pt/Pd alloy structures.
Response: This has been addressed.
- In Eq.(1), the definition of the formation energy is obscure. Since the Pt/Pd bulks have different metal atoms, the formation energy would be much better to be defined with the unit of eV/atom as that in ref. 34.
Response: This has been addressed.
Statement in the manuscript:
“Table 1. Calculated equilibrium lattice constants in angstroms (Å), Volume in cubic angstroms (Å3), and Formation energies of pristine Pt and Pd, as well as the Pt/Pd alloy structures in electron volt per formula unit (f.u).”
Reviewer 2 Report
The paper of Botha et al. reports the results of the DFT study of alloys in the Pd-Pt system. The authors present and discuss electronic, elastic, phonon, and thermodynamic properties calculated with the DFT approach. This information is quite important in view of catalytic applications of these alloys, and the topic matches the scope of the journal. However, it is very difficult to follow the discussion of the results because a lot of information is missing or contradicts the literature data, especially those obtained by experiments. Specifically:
1. Several papers have reported experimental studies of the Pd-Pt binary system. Among those, the paper by Darby and Myles (Metall. Trans. 1972, 3, 653) is the most comprehensive as it reports a detailed investigation of thermodynamics of this system. Specifically, they have shown that Pd and Pt form an unlimited solid solution with the Cu-type crystal structure (Fm-3m). The unit cell parameter of this phase ranges between 3.890(1) Å for Pd and 3.928(1) Å for Pt, with a pronounced negative deviation from the Vegard’s law.
2. Moreover, no information on other phases in the Pd-Pt system is available in the literature. The compounds PtPd3, PtPd, Pt3Pd, and Pt7Pd studied in this paper have never been reported and the respective compositions belong to the Pd-Pt solid solution (Fm-3m). This issue must be discussed.
3. Information presented in Table 1 is incomplete and in part erroneous. For instance, If the lattice parameter “a” of Pt7Pd is 5.550 Å, then the unit volume of the cubic cell (Fm-3m, as reported in section 3.2.2) cannot be 120.81 Å. Secondly, section 3.2.2 tells that PtPd is rhombohedral, space group R-3m; however, Table 1 provides only the “a” parameter, whereas the angle “alpha” is missing; besides, the unit volume of 29.76 Å is unbelievably low for the “a” parameter of 4.735 Å.
4. If the authors are sure in their calculations of the equilibrium crystal structures, they ought to provide all necessary structural information including space groups, unit cell parameters, and atomic coordinates.
5. Finally, in Introduction, the authors announce that “in section 2 the computational methodologies are detailed, followed by the results and discussions in section 3 with conclusions made in section 4”. However, the actual flow of the article is different, with section numbers confused.
Only minor improvement of English language is required.
Author Response
- Several papers have reported experimental studies of the Pd-Pt binary system. Among those, the paper by Darby and Myles (Metall. Trans. 1972, 3, 653) is the most comprehensive as it reports a detailed investigation of thermodynamics of this system. Specifically, they have shown that Pd and Pt form an unlimited solid solution with the Cu-type crystal structure (Fm-3m). The unit cell parameter of this phase ranges between 3.890(1) Å for Pd and 3.928(1) Å for Pt, with a pronounced negative deviation from the Vegard’s law.
Response: This has been addressed with the following statement added to the manuscript.
Statement in the manuscript:
“The lattice constants of alloys either obey Vegard’s law or Retger’s law, but they cannot simultaneously satisfy both laws.[32] Vegard’s law states that the lattice parameters of a continuous substitutional solid solution vary linearly with concentration at a constant temperature when the bonding nature is similar in the constituent phases. On her other hand, Retger’s law states that the mole fraction and cell volume of an alloy have a linear variation with each other. Consequently, alloys with higher mole fraction exhibit greater cell volumes.[32]
In this study, the lattice constants of the Pt3Pd and PtPd3 alloy structures were found to deviate linearly from those of pristine Pt and Pd, thus adhering to Vegard’s law. However, it is important to note that Vegard’s law is an approximation and binary alloys can deviate significantly from it in certain cases.[33–37] The Pt3Pd and PtPd3 alloy structures exhibit similar mole fractions and cell volume (4 atoms and volumes of 59-60 Å3) as the pristine Pt and Pd structures.
Conversely, the Pt7Pd and PtPd alloy structures, with different mole fractions and cell volumes compared to pristine Pt and Pd structures, follows Retger’s law. The Pt7Pd alloys has a mole fractions of 8 atoms and a cell volume of 121 Å3, while the PtPd alloy has a mole fraction of 2 atoms and a cell volume of 30 Å3.
Therefore, the differences in lattice constants and volumes observed in the Pt/Pd alloy structures are a consequence of the varying mole fraction. This indicates that the configuration of Pt/Pd alloys can be properly described by the convex hull method, taking into account the composition-dependent changes in lattice parameters and volumes.”
- Moreover, no information on other phases in the Pd-Pt system is available in the literature. The compounds PtPd3, PtPd, Pt3Pd, and Pt7Pd studied in this paper have never been reported and the respective compositions belong to the Pd-Pt solid solution (Fm-3m). This issue must be discussed.
Response: This has been addressed.
Statement in the manuscript:
“The initial pristine Pt and Pd structures, as well as the Pt/Pd alloy structures, are depicted using the convex hull in Figure 1. The Pt3Pd and PtPd3 alloys are cubic structures and belong to the 221 space group (Pmm). The Pt7Pd alloy structure, on the other hand, is also cubic and belongs to the 225 space group (Fmm). On the other hand, the PtPd alloy structure is trigonal and belongs to the 166 space group (Rm ).“
- Information presented in Table 1 is incomplete and in part erroneous. For instance, If the lattice parameter “a” of Pt7Pd is 5.550 Å, then the unit volume of the cubic cell (Fm-3m, as reported in section 3.2.2) cannot be 120.81 Å. Secondly, section 3.2.2 tells that PtPd is rhombohedral, space group R-3m; however, Table 1 provides only the “a” parameter, whereas the angle “alpha” is missing; besides, the unit volume of 29.76 Å is unbelievably low for the “a” parameter of 4.735 Å.
Response: This has been addressed.
Statement in the manuscript:
“The equilibrium volume is available in the supplementary information (Table S1 and Figure S1), along with the space groups and unit cell parameters”
- If the authors are sure in their calculations of the equilibrium crystal structures, they ought to provide all necessary structural information including space groups, unit cell parameters, and atomic coordinates.
Response: This has been addressed
Statement in the manuscript:
“The equilibrium volume is available in the supplementary information (Table S1), along with the space groups and unit cell parameters”
- Finally, in Introduction, the authors announce that “in section 2 the computational methodologies are detailed, followed by the results and discussions in section 3 with conclusions made in section 4”. However, the actual flow of the article is different, with section numbers confused.
Response: This has been addressed.
Statement in the manuscript:
“This paper is structured as follows: Section 2 presents the results and discussions, followed by a detailed description of the computational methodologies in Section 3. Finally, the conclusions are presented in Section 4.”
Reviewer 3 Report
I have thoroughly reviewed the manuscript titled "Ab-initio study of structural, electronic, and thermal properties of Pt/Pd based alloys". The authors have presented a comprehensive study using Density functional theory (DFT) and have provided valuable insights into the properties of Pt/Pd alloys.
However, there are several areas where the manuscript could be improved to enhance its clarity and reproducibility.
1. To ensure the reproducibility of the work, it would be beneficial if the authors could provide the VASP input files as supporting information. This would allow other researchers to replicate the study and verify the results.
2. The manuscript could benefit from a more detailed explanation of how the elastic constants (Cij), bulk modulus (B), shear modulus (G), Young modulus (E), and Poisson (v) were computed. This would provide readers with a better understanding of the methodologies used in the study.
3. The discussion of formation energies could be expanded. The authors should provide more context and interpretation of these results, as they are crucial for understanding the stability of the alloys.
4. In section 3.2.2. Phonon properties, authors states that "No imaginary modes (negative frequencies) were observed in all the considered Pt/Pd alloy structures. Therefore, the ??3? (Pt3Pd, PtPd3), ??3? (Pt7Pd) and the ?3? (PtPd) phases of the considered Pt/Pd alloy structures are dynamically stable."
However, absence of imaginary frequency only indicates that the structures are local minima. To verify dynamic stability, I suggest authors should perform molecular dynamics simulations.
5. In section 2.2.3. Thermodynamic properties, the relationship equation should be added after
"The relationship between temperature and the Helmholtz free energy (F), entropy (S), and constant-volume specific heat (Cv) is shown in Error! Reference source not found.."
6. It has come to my attention that the PDF file of the manuscript contains instances where "Error! Reference source not found" appears in place of in-text citations. This issue may have occurred during the conversion of the submitted files to PDF format. I kindly request the authors to thoroughly review the PDF file and ensure that all in-text citations are properly formatted and linked to their respective references.
Considering these comments, I recommend that the manuscript be revised before it is considered for publication. The authors should address the aforementioned points and provide a more detailed analysis of their results. This would greatly enhance the quality of the manuscript and make it a significant contribution to the field.
Minor editing is required.
Author Response
- To ensure the reproducibility of the work, it would be beneficial if the authors could provide the VASP input files as supporting information. This would allow other researchers to replicate the study and verify the results.
Response: This has been addressed and added to the supplementary section.
Statement in the manuscript:
“For the purpose of repeatability, the input files utilised in this study have been included in the supplementary information.”
- The manuscript could benefit from a more detailed explanation of how the elastic constants (Cij), bulk modulus (B), shear modulus (G), Young modulus (E), and Poisson (v) were computed. This would provide readers with a better understanding of the methodologies used in the study.
Response: This has been addressed and added to the supplementary section.
Statement in the manuscript:
“A detailed description of the computation process of the elastic constants (Cij), bulk modulus (B), shear modulus (G), Young modulus (E), and Poisson (v) are provided in the supplementary section.”
- The discussion of formation energies could be expanded. The authors should provide more context and interpretation of these results, as they are crucial for understanding the stability of the alloys.
Response: This has been addressed
Statement in the manuscript:
“The formation energy determines the energy change when creating an alloy structure from its constituent elements. Thus, negative formation energies implies that the Pt/Pd alloy structures are energetically stable and likely to form spontaneously.”
- In section 3.2.2. Phonon properties, authors states that "No imaginary modes (negative frequencies) were observed in all the considered Pt/Pd alloy structures. Therefore, the ??3?(Pt3Pd, PtPd3), ??3? (Pt7Pd) and the ?3? (PtPd) phases of the considered Pt/Pd alloy structures are dynamically stable." However, absence of imaginary frequency only indicates that the structures are local minima. To verify dynamic stability, I suggest authors should perform molecular dynamics simulations.
Response: This has been addressed and the required documents were added to the supplementary.
“Crystalline structures exhibit less diffuse pair correlation functions due to the atoms vibrating around high symmetry points. In contrast, liquids have more diffuse pair correlation functions as the average positions of atoms are spread out over a wider range of distances. As the temperature decreases, the pair correlation function becomes more structured, reflecting the ordering of atoms in a crystalline lattice, as shown in Figure S2. These results indicate a high degree of dynamical stability in the pristine Pt and Pd, as well as the Pt/Pd alloy structures, which remain consistent with temperature changes.”
- In section 2.2.3. Thermodynamic properties, the relationship equation should be added after "The relationship between temperature and the Helmholtz free energy (F), entropy (S), and constant-volume specific heat (Cv) is shown in Error! Reference source not found.."
Response: This was addressed, and a detailed description was added to the supplementary information.
Statement in the manuscript:
“As with the elastic properties, a detailed mathematical equation and a comprehensive description of the thermodynamic properties are available in the supplementary information.”
“The relationship between these properties is given in equations (4) – (6), respectively for the Helmholtz free energy (4), entropy (4) and constant-volume specific heat (5).[55] From these equations, the Helmholtz free energy is related to the entropy and temperature (4), whereas the constant-volume specific heat is expressed in terms of the second derivative of the Helmholtz free energy with respect to temperature at a constant volume (6).[55]
(4)
(5)
(6)”
- It has come to my attention that the PDF file of the manuscript contains instances where "Error! Reference source not found" appears in place of in-text citations. This issue may have occurred during the conversion of the submitted files to PDF format. I kindly request the authors to thoroughly review the PDF file and ensure that all in-text citations are properly formatted and linked to their respective references.
Response: This issue has been addressed throughout the manuscript.

Round 2
Reviewer 2 Report
The authors have performed certain correction of their manuscript; however, I’m dissatisfied with their corrections and response.
1. In particular, my second comment “Moreover, no information on other phases in the Pd-Pt system is available in the literature. The compounds PtPd3, PtPd, Pt3Pd, and Pt7Pd studied in this paper have never been reported and the respective compositions belong to the Pd-Pt solid solution (Fm-3m). This issue must be discussed.” Received the following answer:
“The initial pristine Pt and Pd structures, as well as the Pt/Pd alloy structures, are depicted using the convex hull in Figure 1. The Pt3Pd and PtPd3 alloys are cubic structures and belong to the 221 space group (Pmm). The Pt7Pd alloy structure, on the other hand, is also cubic and belongs to the 225 space group (Fmm). On the other hand, the PtPd alloy structure is trigonal and belongs to the 166 space group (Rm ).“
I find this text meaningless because the response simply lists the structures calculated in this paper. However, none of them was found experimentally in several works published previously. My question was about the discussion of the reason for such a discrepancy. Obviously, the authors have nothing to say about it and instead list the structures.
2. My other comment states “Information presented in Table 1 is incomplete and in part erroneous. For instance, If the lattice parameter “a” of Pt7Pd is 5.550 Å, then the unit volume of the cubic cell (Fm-3m, as reported in section 3.2.2) cannot be 120.81 Å. Secondly, section 3.2.2 tells that PtPd is rhombohedral, space group R-3m; however, Table 1 provides only the “a” parameter, whereas the angle “alpha” is missing; besides, the unit volume of 29.76 Å is unbelievably low for the “a” parameter of 4.735 Å.” It received the following explanation by the authors:
“The equilibrium volume is available in the supplementary information (Table S1 and Figure S1), along with the space groups and unit cell parameters”
However, inspecting the corresponding table in SI I found it full of inconsistent information. First, the unit cell dimensions of Pt7Pd do not agree with the space group #225. Second, the metrics given for trigonal PtPd (space group #166) are impossible from the crystallographic point of view. Third, the formula given for calculating the volume of a trigonal structure is totally wrong.
3. My yet another comment was “If the authors are sure in their calculations of the equilibrium crystal structures, they ought to provide all necessary structural information including space groups, unit cell parameters, and atomic coordinates.” The following answer to this comment was:
“The equilibrium volume is available in the supplementary information (Table S1), along with the space groups and unit cell parameters”
The crucial point here is that the authors did not provide atomic coordinates. Apparently, the coordinates are extremely important to check for the interatomic distances and to simulate the corresponding X-ray powder diffraction patterns, because it’s the only way to compare the results of the structure optimization performed in this paper with the experimental results published by other groups.
Therefore, I conclude that the authors have failed to improve their manuscript to the acceptable level, which leads to my recommendation to REJECT this paper.
English is fine
Author Response
Please see the attached document for the response accompanies with figures.
- In particular, my second comment “Moreover, no information on other phases in the Pd-Pt system is available in the literature. The compounds PtPd3, PtPd, Pt3Pd, and Pt7Pd studied in this paper have never been reported and the respective compositions belong to the Pd-Pt solid solution (Fm-3m). This issue must be discussed.” Received the following answer:
“The initial pristine Pt and Pd structures, as well as the Pt/Pd alloy structures, are depicted using the convex hull in Figure 1. The Pt3Pd and PtPd3 alloys are cubic structures and belong to the 221 space group (Pmm). The Pt7Pd alloy structure, on the other hand, is also cubic and belongs to the 225 space group (Fmm). On the other hand, the PtPd alloy structure is trigonal and belongs to the 166 space group (Rm ).“
I find this text meaningless because the response simply lists the structures calculated in this paper. However, none of them was found experimentally in several works published previously. My question was about the discussion of the reason for such a discrepancy. Obviously, the authors have nothing to say about it and instead list the structures.
Response:
In this study, Pt/Pd alloy structures were predicted using the convex hull method. Previous first-principle studies have shown that the convex hull method is reliable in identifying stable structures. [1-6] The feasibility of the realization of the various identified alloys was evaluated considering their formation energy, elastic, thermodynamic, and electronic properties.
[1] Botha, Louise M., et al. "Mixing thermodynamics and electronic structure of the Pt 1− x Ni x (0≤ x≤ 1) bimetallic alloy." RSC advances 9.30 (2019): 16948-16954.
[2] Levy, Ohad, et al. "The new face of rhodium alloys: revealing ordered structures from first principles." Journal of the American Chemical Society 132.2 (2010): 833-837.
[3] Levy, Ohad, Gus LW Hart, and Stefano Curtarolo. "Hafnium binary alloys from experiments and first principles." Acta Materialia 58.8 (2010): 2887-2897.
[4] Seko, A., Shitara, K., & Tanaka, I. (2014). Efficient determination of alloy ground-state structures. Physical Review B, 90(17). doi:10.1103/physrevb.90.174104
[5] Wang, Haidi, et al. "Crystal structure prediction of binary alloys via deep potential." Frontiers in Chemistry 8 (2020): 589795.
[6] Widom, M., and M. Mihalkovic. "Stability of Fe-based alloys with structure type C6Cr23." Journal of materials research 20.1 (2005): 237-242.
- My other comment states “Information presented in Table 1 is incomplete and in part erroneous. For instance, If the lattice parameter “a” of Pt7Pd is 5.550 Å, then the unit volume of the cubic cell (Fm-3m, as reported in section 3.2.2) cannot be 120.81 Å. Secondly, section 3.2.2 tells that PtPd is rhombohedral, space group R-3m; however, Table 1 provides only the “a” parameter, whereas the angle “alpha” is missing; besides, the unit volume of 29.76 Å is unbelievably low for the “a” parameter of 4.735 Å.” It received the following explanation from the authors:
“The equilibrium volume is available in the supplementary information (Table S1 and Figure S1), along with the space groups and unit cell parameters”
However, inspecting the corresponding table in SI I found it full of inconsistent information.
First, the unit cell dimensions of Pt7Pd do not agree with the space group #225.
Response:
In the previous manuscript, the primitive cell of the Pt7Pd structure was used. In the revised manuscript, the primitive cell was replaced with the conventional cell, which agrees with space group #225. See attached document for figures.
Second, the metrics given for trigonal PtPd (space group #166) are impossible from the crystallographic point of view.
Response:
In the previous manuscript, the primitive cell of the PtPd structure was used. In the revised manuscript, the primitive cell was replaced with the conventional cell, which agrees with space group #166. See attached file for the figures.
Third, the formula given for calculating the volume of a trigonal structure is totally wrong.
Response:
The formula was corrected. Refer to supplementary section, Figures S1 and S2 for volume calculation. See attached document for figures.
- The crucial point here is that the authors did not provide atomic coordinates. Apparently, the coordinates are extremely important to check for the interatomic distances and to simulate the corresponding X-ray powder diffraction patterns, because it’s the only way to compare the results of the structure optimization performed in this paper with the experimental results published by other groups.
Response:
The atomic coordinates are available in Supplementary Section 5, Table S1.

Reviewer 3 Report
The authors have adequately addressed the concerns I raised in my initial review of the manuscript titled "Ab-initio study of structural, electronic, and thermal properties of Pt/Pd based alloys". The revised manuscript is much improved, and I recommend it for publication.
Minor proofreading is required.
Author Response
Reviewer 3:
The authors have adequately addressed the concerns I raised in my initial review of the manuscript titled "Ab-initio study of structural, electronic, and thermal properties of Pt/Pd based alloys". The revised manuscript is much improved, and I recommend it for publication.
Minor proofreading is required.
Response: This has been addressed.